# Mitochondrial Permeability Transition Causes Mitochondrial Reactive Oxygen Species- and Caspase 3-Dependent Atrophy of Single Adult Mouse Skeletal Muscle Fibers

**DOI:** 10.3390/cells10102586

**Published:** 2021-09-29

**Authors:** Sarah K. Skinner, Angelo Solania, Dennis W. Wolan, Michael S. Cohen, Terence E. Ryan, Russell T. Hepple

**Affiliations:** 1Department of Physical Therapy, University of Florida, Gainesville, FL 32610, USA; sk.skinner@ufl.edu; 2Departments of Molecular Medicine and Integrative Structural and Computational Biology, Scripps Research, La Jolla, San Diego, CA 92037, USA; asolania@scripps.edu (A.S.); wolan@scripps.edu (D.W.W.); 3Department of Chemical Physiology and Biochemistry, Oregon Health and Science University, Portland, OR 97239, USA; cohenmic@ohsu.edu; 4Department of Applied Physiology and Kinesiology, University of Florida, Gainesville, FL 32610, USA; ryant@ufl.edu

**Keywords:** mitochondrial permeability transition pore, ROS, caspase-3, skeletal muscle atrophy

## Abstract

Elevated mitochondrial reactive oxygen species (mROS) and an increase in caspase-3 activity are established mechanisms that lead to skeletal muscle atrophy via the upregulation of protein degradation pathways. However, the mechanisms upstream of an increase in mROS and caspase-3 activity in conditions of muscle atrophy have not been identified. Based upon knowledge that an event known as mitochondrial permeability transition (MPT) causes an increase in mROS emission and the activation of caspase-3 via mitochondrial release of cytochrome c, as well as the circumstantial evidence for MPT in some muscle atrophy conditions, we tested MPT as a mechanism of atrophy. Briefly, treating cultured single mouse flexor digitorum brevis (FDB) fibers from adult mice with a chemical inducer of MPT (Bz423) for 24 h caused an increase in mROS and caspase-3 activity that was accompanied by a reduction in muscle fiber diameter that was able to be prevented by inhibitors of MPT, mROS, or caspase-3 (*p* < 0.05). Similarly, a four-day single fiber culture as a model of disuse caused atrophy that could be prevented by inhibitors of MPT, mROS, or activated caspase-3. As such, our results identify MPT as a novel mechanism of skeletal muscle atrophy that operates through mROS emission and caspase-3 activation.

## 1. Introduction

Skeletal muscle atrophy occurs in a wide variety of clinical conditions including aging, neuromuscular diseases, lung disease, cardiovascular disease, kidney disease, and cancer, and is associated with exacerbated health outcomes [1,2,3,4,5,6]. A better understanding of the pathways underlying muscle atrophy is essential to develop and implement effective treatment strategies aimed at preserving muscle mass to improve health outcomes in these clinical conditions. To date, studies have identified an elevated mitochondrial reactive oxygen species (mROS) [7] and an increase in caspase-3 activity as mechanisms involved in skeletal muscle atrophy [8]. Specifically, mROS activates transcription factors, such as FoxOs 1, 3, and 4, which promote muscle atrophy through the upregulation of both autophagy genes (including LC3 and Bnip3) [9] and ‘atrogenes’ like Mafbx/atrogin-1, MuRF-1, and Fbxl22 which ubiquitinate proteins for subsequent degradation by the 26S proteasome [10,11]. Similarly, activated caspase-3 promotes muscle atrophy by cleaving actin from muscle cross-bridges, making it available for subsequent ubiquitination [12], and cleaves negative regulators of the 19S proteasome to allow for an increase in proteasome activity [13]. Despite this knowledge, the mechanisms responsible for the increases in mROS and caspase-3 activity in the context of muscle atrophy have not been fully resolved.

Under certain circumstances, mitochondria are known to undergo an event known as mitochondrial permeability transition (MPT), in which a non-specific pore forms across the inner mitochondrial membrane [14]. This allows for the passage of mitochondrial matrix constituents into the intermembrane space, causes collapse of the mitochondrial membrane potential, and causes mitochondrial swelling that can rupture the outer mitochondrial membrane [15,16]. This releases both mROS and mitochondrial proteins, such as the caspase-3 activator cytochrome c, into the cytoplasm where they can activate their associated signaling cascades. Despite MPT being associated with both an increase in mROS and caspase-3 activity, prior studies have not considered the possibility of MPT causing skeletal muscle atrophy.

Although MPT was not directly tested as a mechanism of atrophy in prior works, several studies have provided evidence of MPT occurring in skeletal muscle in conditions associated with atrophy [1,17,18]. Previous work from our group noted that skeletal muscle atrophy in healthy, physically active septuagenarian men was associated with a greater nuclear co-localization of Endonuclease G, a mitochondrial protein released during MPT that induces nuclear DNA degradation, compared to similarly active young adult men [1]. Furthermore, permeabilized muscle bundles from these septuagenarian men evidenced a reduced time to MPT in response to a Ca^2+^ challenge compared to young adult men, consistent with mitochondria being sensitized to MPT in atrophying aging human muscle [1]. Similarly, Powers et al. noted that following 14 days of skeletal muscle disuse induced by hindlimb suspension, rat soleus muscle mitochondria demonstrate organelle swelling and distorted cristae [17], which are established morphological features of mitochondria that have undergone MPT [19] and appear to be related to the formation of Ca^2+^-phosphate granules [20]. Moreover, a model of denervation muscle atrophy involving sciatic nerve transection reduces skeletal muscle mitochondrial time to Ca^2+^-induced MPT [18], and causes an increase in so-called mito-flash activity consistent with an increase in MPT events [21]. Despite this evidence, to our knowledge, no direct assessment of the relationship between MPT and skeletal muscle atrophy has previously been conducted.

Therefore, to address the role of MPT in skeletal muscle atrophy, we examined the impact of MPT on isolated single mouse flexor digitorum brevis (FDB) muscle fibers. Specifically, we determined the effect of the acute chemical induction of MPT on mROS emission and caspase-3 activity, as well as how co-treatment with an inhibitor of MPT, mROS, or caspase-3 modulated changes in the myofiber diameter over this same time period. Finally, we used a multi-day cultured single fiber model of muscle ‘disuse’ atrophy to determine if reducing MPT, mROS, or caspase-3 activity could prevent myofiber atrophy in this setting. We hypothesized that acute chemical induction of MPT would cause an increase in mROS and caspase-3 activity in association with muscle fiber atrophy, whereas an inhibition of MPT, mROS or caspase-3 would prevent these changes. We also hypothesized that the inhibition of MPT, mROS or caspase-3 in a single fiber model of muscle ‘disuse’ would prevent myofiber atrophy. The overall scheme of how we hypothesize that MPT integrates with mROS and caspase-3 in causing muscle atrophy is graphically represented in Figure 1.

## 2. Materials and Methods

### 2.1. Animals and Surgical Methods

All procedures were conducted with approval from the University of Florida Institutional Animal Care and Use Committee (protocol #202011171, to R. Hepple). Male C57BL/6 mice were housed singly in ventilated cages and provided food and water ad libitum. On the day of sacrifice, mice were anesthetized with 2–4% isoflurane and the FDBs were carefully dissected and removed from both hindlimbs, trimmed of excess fat, blood vessels, and connective tissue in room-temperature physiological rodent saline (PRS: 138 mM NaCl, 2.7 mM KCl, 1.8 mM CaCl_2_, 1.06 mM MgCl_2_, 12.4 mM HEPES and 5.6 mM glucose, pH 7.3). Mice were killed by thoracotomy and removal of the heart following isolation of the muscles.

### 2.2. Single Fiber Isolation and Culture

Following the removal of non-muscle tissue, FDBs were prepared for single fiber isolation according to the method of Komiya and colleagues [22]. Briefly, FDBs were incubated in an Eppendorf tube containing collagenase solution (1.5 mL PRS containing 0.2% collagenase type I (Sigma-Aldrich, St. Louis, MO, USA), 0.1% elastase (Sigma-Aldrich), 0.0625% protease from *Streptomyces griseus* (Sigma-Aldrich), 0.033% dispase (Invitrogen, Waltham, MA, USA), and 10% fetal bovine serum (FBS, Invitrogen)) at 37 °C and 5% CO_2_ for 90 min to digest the remaining connective tissue. After digestion, muscles were moved to a 35 mm culture dish containing proliferative medium (PM: high-glucose Dulbecco’s modified Eagle’s medium (DMEM; Life Technologies, Carlsbad, CA, USA) containing 10% FBS, 1% antibiotic-antimycotic mix (Life Technologies), and 0.1% gentamycin (Life Technologies) using a wide-bore Pasteur pipette. The muscles were then triturated with a P1000 pipette for dissociation into single muscle fibers. Every 15 min, fibers were placed back at 37 °C and 5% CO_2_ for 5 min to avoid substantial change in media temperature. The fibers were transferred into a 5 mL glass centrifuge tube containing 10 mL of PM for a gravity sedimentation wash, which was repeated a second time. PM was then carefully removed, and the muscle fibers were re-suspended in 1 mL of maintenance medium (MM: high-glucose DMEM supplemented with 20% 1X Serum Replacement 2 (Sigma-Aldrich)). Individual muscle fibers were transferred to 35 mm culture dishes (*n* = 25–50 fibers for 24 h experiments; *n* = 50–100 fibers for multi-day incubation disuse experiments) containing MM and kept under standard culture conditions for the duration of the experiment. All muscle fibers were imaged by brightfield illumination on a customized Leica SP8 microscope (Leica Microsystems, Wetzlar, Germany) following plating to permit baseline measurements of myofiber diameter. Briefly, culture dishes were placed in a pre-warmed stage-top incubator (Tokai Hit, Shizuoka, Japan) at 37 °C, and tiled images were taken of the entire dish. The images were stitched and exported for analysis in ImageJ. This process was repeated for each dish at the conclusion of each experiment to determine changes in the myofiber diameter. Ten diameter measurements were collected along the length of each fiber and averaged to provide a mean fiber diameter value. Fibers that were visually hypercontracted were excluded from the analysis (Appendix A). To confirm that changes in fiber length did not bias diameter measurements, fiber length was assessed at baseline, following one day, and following four days of culture conditions in the multi-day culture model (Appendix A). There was no change in myofiber length from baseline over the four-day culture period, indicating that changes in fiber length are unlikely to impact muscle fiber diameter measurements.

### 2.3. mROS Emission and Caspase-3 Activity

To determine the impact of chemical induction of MPT on mROS emission and caspase-3 activation, Bz423 (Bio-Techne Corporation, Minneapolis, MN, USA) was used to chemically induce MPT in cultured single living myofibers. Bz423 is an inhibitor of ATP synthase [23] that favors MPT pore opening by interacting with the oligomycin sensitivity conferring protein (OSCP) subunit of the F0-F1 ATP synthase [24]. For all experiments using Bz423, a concentration of 300 nM was used as has been done previously to induce MPT in cells [25], and was sufficient to cause atrophy in our experiments. A higher concentration of 6 µM Bz423 was also tested, and though it also induced atrophy, this concentration appeared to cause more fiber death over the treatment period (S. Skinner, unpublished observations). Briefly, for mROS experiments, fibers were treated for 24 h with 300 nM Bz423 alone, 300 nM + 5 µM mitoTEMPO (to scavenge mROS), or 300 nM Bz423 + TR002 (to inhibit MPT). For experiments assessing caspase-3 activation, muscle fibers were treated for 24 h with 300 nM Bz423, 300 nM Bz423 + 20 µM Ac-ATS010-KE (a novel, selective, rapid-cell permeable inhibitor of caspase-3 [26]), or 300 nM Bz423 + 1 µM TR002.

mitoSOX (a live cell-permeable fluorescent indicator of mitochondrial superoxide; Sigma Aldrich) and caspase-3 FLICA (fluorescent probe targeted to activated caspase-3; ImmunoChemistry Technologies LLC, Bloomington, MN, USA) were used in combination with the acute treatments listed above. Briefly, mitoSOX was added to MM for a final concentration of 0.1 µM and fibers were incubated for 30 min under standard culture conditions, while for activated caspase-3 experiments, FLICA reagent was added 1:30 to MM and fibers were incubated for 1 h under the same conditions. Then, MM was carefully removed, and the fibers were rinsed 3× to remove excess reagent and any residual phenol red from the MM; fibers treated with mitoSOX were rinsed in phosphate-buffered saline (PBS), whereas fibers treated with caspase-3 FLICA were rinsed using 1x Apoptosis Wash Buffer (ImmunoChemistry Technologies, LLC). Next, fibers were imaged using a brightfield camera and TexasRed filter cube, then imaged again using brightfield only for confirmation of fiber viability during analysis. As mitoSOX and caspase-3 FLICA bind irreversibly to their respective targets, one dish of myofibers was used to determine baseline fluorescence, while separate dishes were imaged at baseline in brightfield only, and then the respective treatment was added, and fibers were kept in standard culture conditions for 24 h. Following the 24 h treatment, mitoSOX or caspase-3 FLICA were added to the appropriate dishes and imaged in fluorescence and brightfield as described above. All images were stitched and exported for quantification of fluorescent signal using ImageJ. The fluorescent area of each fiber was outlined using the freehand drawing tool, and integrated density measures were analyzed. Fibers that were hypercontracted (ratio of length: mean myofiber diameter < 5) were excluded from analyses. Integrated density values (product of area and mean gray value) were then compared between baseline and 24 h to determine change in mROS emission or caspase-3 activation. Representative images of baseline mitoSOX and caspase-3 FLICA fluorescence may be found in Appendix A.

### 2.4. Acute Inhibition of MPT, mROS and Caspase-3 Activity

To inhibit MPT in single fiber experiments, we used 1 μM of TR002, which is a novel, small molecule inhibitor of MPT that acts independently of cyclophilin D [27]. Myofiber diameter was measured at baseline and following 24 h Bz423 treatment, as described above.

Acute experiments were also conducted to determine if inhibiting mROS or caspase-3 activation prevented atrophy following treatment with Bz423. Myofibers were isolated and plated as described above, imaged, then treated for 24 h with 300 nM Bz423 (to induce MPT) or co-treated with 300 nM Bz423 and 1 µM TR002 (to inhibit MPT), 300 nM Bz423 + 5 µM mitoTEMPO (mitochondrially-targeted antioxidant (Sigma-Aldrich)), or 300 nM Bz423 + 20 µM Ac-ATS010-KE (inhibitor of caspase-3). The fibers were imaged again 24 h after treatment, and images were stitched and exported to ImageJ for analysis of change in myofiber diameter, as described above.

### 2.5. ‘Disuse’ Atrophy Experiments

Muscle disuse models, such as ankle-joint immobilization and hindlimb suspension, are associated with a substantial decline in skeletal muscle size [28]. Similarly, isolated skeletal muscle fibers that are cultured for an extended period of time (4–5 days) can be used as a model of muscle disuse, as they are no longer innervated and are no longer activated to contract [29]. To look at the effects of preventing MPT, mROS, or caspase-3 activation on myofiber diameter in this model of muscle ‘disuse,’ myofibers were plated and treated with either DMSO (vehicle control), 1 µM TR002 (to inhibit MPT), 5 µM mitoTEMPO (to scavenge mROS), or 20 µM Ac-ATS010-KE (to inhibit caspase-3) for four days under standard culture conditions. The media was carefully changed every two days, and fibers were imaged at baseline and at the end of the four-day treatment period. The images were exported to ImageJ and myofiber diameter was determined at baseline and post-treatment as described above.

A similar multi-day culture experiment was also conducted wherein myofibers were treated for five days with either DMSO or 1 µM cyclosporine A (CsA; Sigma-Aldrich), a more widely used MPT inhibitor that acts through inhibition of the MPT-regulating protein, cyclophilin D [30]. The media was changed every two days, and the fibers were imaged at baseline and following the five-day treatment period. Image analysis was conducted as described above.

### 2.6. Statistics

To compare changes in myofiber diameter from pre- to post-treatment in acute myofiber culture experiments, nested t-tests or nested one-way ANOVAs were used to account for multiple technical replicates from individual animals (biological replicates). For mROS and caspase-3 fluorescent analyses, Brown–Forsythe ANOVA tests were used to compare fluorescent signal values pre- to post-treatment. To analyze change in myofiber diameter in disuse atrophy experiments, nested one-way ANOVA was used. The values are presented as mean ± SD. For all statistical analyses we set α = 0.05.

## 3. Results

### 3.1. Acute Myofiber Culture Experiments

Single myofibers treated for 24 h with 300 nM Bz423 to induce MPT demonstrated significantly greater mitoSOX fluorescence signal after 24 h treatment compared to pre-treatment values (mitoSOX: 42,038 ± 42,272 vs. 80,453 ± 51,720 AU, *p* < 0.01 (Figure 2), indicating that chemical induction of MPT increases mROS. This increase in mitoSOX fluorescence was prevented by treatment with either 5 µM mitoTEMPO (to scavenge mROS; 42,038 ± 42,272 vs. 41,611 ± 28,609 AU, *p* > 0.999) or 1 µM TR002 (to inhibit MPT; 42,038 ± 42,272 vs. 31,682 ± 13,370 AU, *p* = 0.484). Similarly, myofibers treated for 24 h with 300 nM Bz423 demonstrated significantly greater caspase-3 FLICA fluorescence signal following the treatment period compared to baseline (99813 ± 77,813 vs. 244,343 ± 124,166 AU, *p* < 0.0001) (Figure 3), indicating an increase in caspase-3 activity. Treatment with 20 µM Ac-ATS010-KE (to inhibit caspase-3) caused a decrease in caspase-3 FLICA signal compared to the baseline (99813 ± 77,813 vs. 22,968 ± 21,491 AU, *p* < 0.0001) while myofibers treated with 1 µM TR002 in combination with Bz423 demonstrated no change in caspase-3 FLICA signal compared to baseline (99813 ± 77,813 vs. 59,851 ± 40,704 AU; *p* = 0.108). Moreover, 24 h treatment with 300 nM Bz423 led to ~20% decrease in myofiber diameter from pre- to post-treatment (38.05 ± 7.104 vs. 30.41 ± 8.975 µm, *p* = 0.021), indicating that chemical induction of MPT causes atrophy in isolated single mouse muscle fibers (Figure 4). However, fibers treated with 300 nM Bz423 in combination with 1 µm TR002 (MPT inhibitor) showed no change in myofiber diameter over this same period (35.02 ± 8.472 vs. 35.43 ± 7.895 µm, *p* = 0.979), demonstrating that the inhibition of MPT prevents atrophy in this model. Similarly, when myofibers were treated for 24 h with 300 nM Bz423 + 5 µM mitoTEMPO (to inhibit mROS) or 300 nM Bz423 + 20 µM Ac-ATS010-KE (to inhibit caspase-3), they showed no change in diameter from pre- to post-treatment (mitoTEMPO: 39.43 ± 6.886 vs. 38.27 ± 6.935 µm, *p* > 0.05; Ac-ATS010-KE: 37.61 ± 8.059 vs. 36.52 ± 7.309 µm, *p* > 0.05), whereas myofibers treated with 300 nM Bz423 again demonstrated a significant decrease in diameter (39.75 ± 10.25 vs. 32.25 ± 7.513 µm, *p* = 0.04) (Figure 5). Collectively, our results demonstrate that the acute chemical induction of MPT causes atrophy and increases mROS and caspase-3 activity in isolated single mouse muscle fibers, while the inhibition of any one of MPT, mROS, or caspase-3 prevents the atrophy noted in this model.

### 3.2. ‘Disuse’ Muscle Atrophy Model Experiments

Multi-day incubation of isolated single myofibers has been shown to result in a decrease in myofiber size, which allows for the use of this system as a model of muscle ‘disuse’ atrophy [29]. Therefore, to test if preventing MPT, mROS, or caspase-3 activation prevents atrophy in this model of muscle disuse, we cultured isolated single myofibers for four days while treating with DMSO (vehicle control), 1 µM TR002 (to inhibit MPT), 5 µM mitoTEMPO (to inhibit mROS), or 20 µM Ac-ATS010-KE (to inhibit caspase 3). Whereas myofibers treated with DMSO showed a significant decline in diameter from baseline to day 4 (37.18 ± 7.330 vs. 31.51 ± 8.678 µm, *p* = 0.014), myofibers treated with TR002, mitoTEMPO, or Ac-ATS010-KE demonstrated no change in diameter from pre- to post-treatment (TR002: 37.55 ± 7.806 vs. 35.61 ± 8.086 µm, *p* = 0.814; mitoTEMPO: 35.29 ± 8.742 vs. 35.41 ± 8.024 µm, *p* > 0.999; Ac-ATS010-KE: 37.62 ± 7.178 vs. 37.61 ± 8.313, *p* > 0.999) (Figure 6). These results therefore indicate that the inhibition of MPT, mROS, or caspase-3 prevents muscle atrophy in an isolated single mouse muscle fiber model of ‘disuse’ atrophy. Importantly, we also performed a multi-day single muscle fiber incubation experiment using cyclosporine A (CsA), which is a more widely used inhibitor of MPT that operates through the inhibition of the MPT-regulating protein cyclophilin D [30]. Similar to our results with TR002, CsA also prevented the atrophy occurring with multi-day maintenance of muscle fibers in culture conditions (Appendix A).

## 4. Discussion

The purpose of this study was to test the relationship between MPT and skeletal muscle atrophy in single living mouse muscle fibers, as well as to determine if mROS emission and caspase-3 activation mediate this relationship. We hypothesized that an acute chemical induction of MPT would increase mROS and caspase-3 activation and induce atrophy in single mouse FDB fibers, while the inhibition of MPT, mROS, or caspase-3 would prevent this atrophy. Specifically, we found that 24 h treatment of single fibers with Bz423, a chemical inducer of MPT, was associated with an increase in mROS emission, as measured by mitoSOX fluorescence, and an increase in activated caspase-3, as determined by a quantification of the FLICA signal. Consistent with our hypothesis, Bz423 treatment also caused a 20% decrease in myofiber diameter. Moreover, the inhibition of MPT, mROS, and caspase-3 each prevented atrophy in single muscle fibers following acute treatment with Bz423, indicating that the MPT-induced atrophy in this model is mediated by mROS emission and caspase-3 activation. This latter finding shows that MPT-induced atrophy operates through established mechanisms seen in other atrophy conditions. We also explored the role of MPT in the atrophy occurring in a single fiber model of disuse [29]. Consistent with a role for MPT in disuse atrophy involving mROS and caspase-3, fibers treated with vehicle control demonstrated atrophy after four days of culture, whereas fibers treated with inhibitors of any one of the MPT, mROS, or caspase-3, showed no change in myofiber diameter over this same period. Therefore, our results identify MPT as a novel mechanism of skeletal muscle atrophy that operates through both mROS emission and caspase-3 activation.

### 4.1. mROS Emission and Caspase-3 Activation Increase Following MPT in Single Isolated Mouse Muscle Fibers, and Are Accompanied by a Decrease in Myofiber Diameter

Although MPT plays a known pathological role in ischemic organ damage through mROS, mitochondrial Ca^2+^ release, and the activation of intrinsic apoptotic pathways [31], and evidence of MPT has been observed in aged skeletal muscle that demonstrates atrophy [1], a direct examination of the role of MPT in mechanisms of muscle atrophy has not previously been conducted. MPT occurs in response to various stressors, including ROS and Ca^2+^ overload, and involves the formation of a non-specific pore known as the mitochondrial permeability transition pore (mPTP) across the inner mitochondrial membrane. Although research continues, current evidence suggests that the formation of the mPTP involves a Ca^2+^-dependent conformational change of the F_o_F_1_-ATP synthase to form the channel [32]. Furthermore, there is also evidence in patch-clamped mitochondrial membranes that mPTP activity requires the presence of at least one of the isoforms of the adenine nucleotide translocase (also known as the ADP/ATP carrier) [33], and recent work suggests the ANT may contribute to channel formation in the absence of an assembled F-ATP synthase [34]. A prolonged opening of the mPTP results in a collapse of the mitochondrial membrane potential, and can cause rupture of the outer mitochondrial membrane, leading to the release of both mROS and mitochondrial proteins, such as apoptosis inducing factor (AIF), Endonuclease G (EndoG), and cytochrome c [35]. Both AIF and EndoG can travel to the nucleus to carry out chromatin condensation and DNA fragmentation [36,37], while mROS and cytochrome c can act as signaling molecules. Specifically, mROS is able to activate transcription factors, such as the FOXO family, which are associated with the upregulation of ‘atrogenes’ like the E3-ubiquitin ligases *Mafbx*/*atrogin-1* and *MuRF-1*, as well as autophagy-promoting genes, such as *LC3*, *Beclin-1*, and *BNIP3* [38]. Similarly, cytochrome c is a component of the apoptosome which ultimately leads to the cleavage of caspase-3 [39]. In turn, activated caspase-3 is involved in the upregulation of proteasome activity through cleavage of negative regulators of the 19S proteasome [13], and is also able to cleave actin from skeletal muscle cross-bridges, allowing for ubiquitination of actin and subsequent degradation by the ubiquitin-proteasome system [12]. As both an increase in mROS and caspase-3 activity have been previously shown to be involved in mechanisms of muscle atrophy, we used a single isolated adult muscle fiber approach to help us test if MPT induces an increase in mROS emission and caspase-3 activation in skeletal muscle, and if this leads to muscle atrophy.

To induce MPT, we used Bz423, a benzodiazepine that acts on the OCSP subunit of the ATP-synthase to promote the open conformation of the mPTP [23]. To examine changes in mROS and caspase-3 activity, we used a mitochondrial targeted fluorescent ROS probe (mitoSOX) and a fluorescent activated caspase-3 probe (caspase-3 FLICA) in isolated mouse muscle fibers at baseline and following 24 h of treatment with Bz423. Relative levels of mROS and caspase-3 activation increased substantially over the Bz423 treatment period, as determined by fluorescent signal quantification, confirming that MPT induces both an increase in mROS emission and an activation of caspase-3 in skeletal muscle fibers. Treatment with TR002 prevented an increase in both mitoSOX and caspase-3 FLICA signal over this same period, indicating that inhibition of MPT prevents the increase in mROS and caspase-3 activity. Similarly, treatment with mitoTEMPO (mROS-scavenger) prevented an increase in mitoSOX signal and Ac-ATS010-KE (caspase-3 inhibitor) reduced caspase-3 FLICA signal in myofibers treated for 24 h with Bz423, confirming the specificity of the mROS and caspase-3 fluorescence methods, respectively. Muscle fibers treated with Bz423 for 24 h also demonstrated a reduced myofiber diameter that was prevented by an MPT inhibitor that is posited to directly block the mPTP (TR002) [27], showing that Bz423-induced atrophy depends on MPT.

To determine if either or both an increase in mROS emission and caspase-3 activity were directly involved in the development of the myofiber atrophy observed following acute chemical induction of MPT, we again treated isolated FDB muscle fibers with Bz423 for 24 h along with either the mitochondrial-targeted antioxidant mitoTEMPO, or a novel inhibitor of caspase-3, Ac-ATS010-KE. Consistent with our hypothesis, treatment with an inhibitor of either mROS or caspase-3 prevented MTP-induced atrophy. As such, our results show that the chemical induction of MPT induces atrophy that is dependent upon mROS and caspase-3 activity, suggesting that MPT operates to recruit established mechanisms of muscle atrophy.

### 4.2. MPT, mROS, and Caspase-3 in a Single Muscle Fiber Model of ‘Disuse’

Although there is indirect evidence of MPT occurring in muscle disuse atrophy, based upon the appearance of mitochondria that are swollen and/or have disrupted cristae following hindlimb suspension [17], the involvement of MPT in the development of disuse atrophy has not been directly examined. Therefore, to further examine the involvement of MPT in the development of atrophy in muscle disuse, we employed a multi-day incubation model using single mouse muscle fibers. Specifically, we kept isolated FDB muscle fibers in culture conditions for four days, as a multi-day culture of single muscle fibers has been previously demonstrated to induce atrophy [29]. To determine the role of MPT in the atrophy seen in this model, we treated muscle fibers with TR002 (to inhibit MPT) for the duration of the experiment and again determined change in myofiber diameter over the four-day period. In agreement with the results of our acute experiments, muscle fibers treated with an inhibitor of MPT showed no change in myofiber diameter from baseline to day 4, indicating that the atrophy occurring in this model of disuse requires MPT. To establish whether the atrophy in this model also depended upon mROS and caspase-3, we also treated groups of fibers with either a mitochondrial-targeted antioxidant or caspase-3 inhibitor for the duration of the four-day incubation. Similar to what was observed with chemical induction of MPT, inhibiting mROS or caspase-3 prevented the muscle fiber atrophy from occurring with this model of muscle disuse. As such, collectively, our results identify MPT as a novel mechanism of muscle atrophy that is upstream of established mechanisms of muscle atrophy involving mROS and caspase-3. This is notable, as up to this point, strategies for preventing muscle atrophy have focused on inhibiting atrogene signaling [40], preventing mROS [7], or preventing caspase-3 activation [8] individually. If all of these are consequent to MPT, as our data suggest, therapeutically targeting MPT could arguably offer a greater overall effect in preserving muscle mass and function by preventing the activation of multiple atrophy mechanisms simultaneously.

Our experiments are not without limitations. In particular, the use of a culture system restricts the conclusions we are able to make regarding the role of MPT in skeletal muscle in vivo. However, by using adult mouse muscle fibers we are using a system that can avoid some of the limitations associated with the less differentiated characteristics of more traditional cell culture systems that use myotubes grown from muscle cell lines [41]. In turn, this allows us to gain insight to the impact of MPT on skeletal muscle in a model that maintains the fully differentiated muscle state seen in vivo. As such, our experiments lay the groundwork for future in vivo experiments aimed at testing the therapeutic potential of targeting MPT in preventing the muscle atrophy seen in classical atrophy models such as hindlimb unloading, denervation, and fasting.

## 5. Conclusions

The purpose of our study was to test MPT as a novel mechanism of muscle atrophy operating through increased mROS emission and caspase-3 activation. To meet this goal, we employed two models involving single isolated adult mouse muscle fibers: (1) acute chemical induction of MPT, and (2) multi-day incubation of single muscle fibers in culture conditions. We demonstrated that in single muscle fibers, the chemical induction of MPT caused an increase in mROS emission and activated caspase-3, and reduced myofiber diameter over a 24 h treatment period. Moreover, we determined that inhibition of mROS, caspase-3, or MPT prevented myofiber atrophy during an acute setting, as well as during a multi-day culture model of muscle disuse. The results of our study therefore identify MPT as a novel mechanism of skeletal muscle atrophy that is upstream of the established mechanisms of muscle atrophy: mROS emission and the activation of caspase-3. In so doing, our results lay the foundation for examining the impact of MPT-inhibition in a variety of preclinical models associated with muscle atrophy.

## Figures and Tables

**Figure 1 cells-10-02586-f001:**
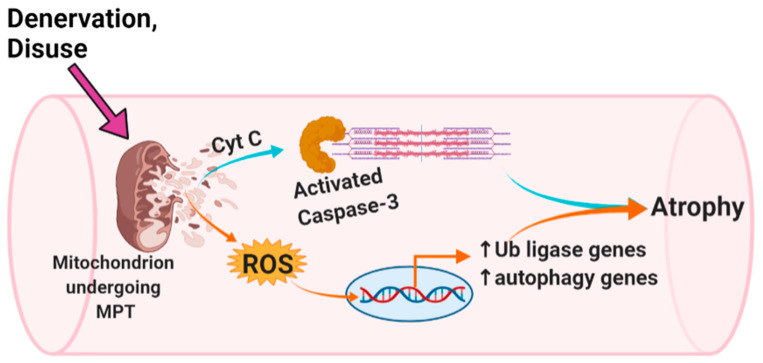
Hypothesized mechanism of mitochondrial permeability transition (MPT)-induced skeletal muscle atrophy. In this scheme, MPT causes atrophy by recruiting previously established mechanisms of muscle atrophy involving caspase 3 activation and mitochondrial ROS-induced upregulation of atrogenes and autophagy genes. This figure was created using Biorender.com.

**Figure 2 cells-10-02586-f002:**
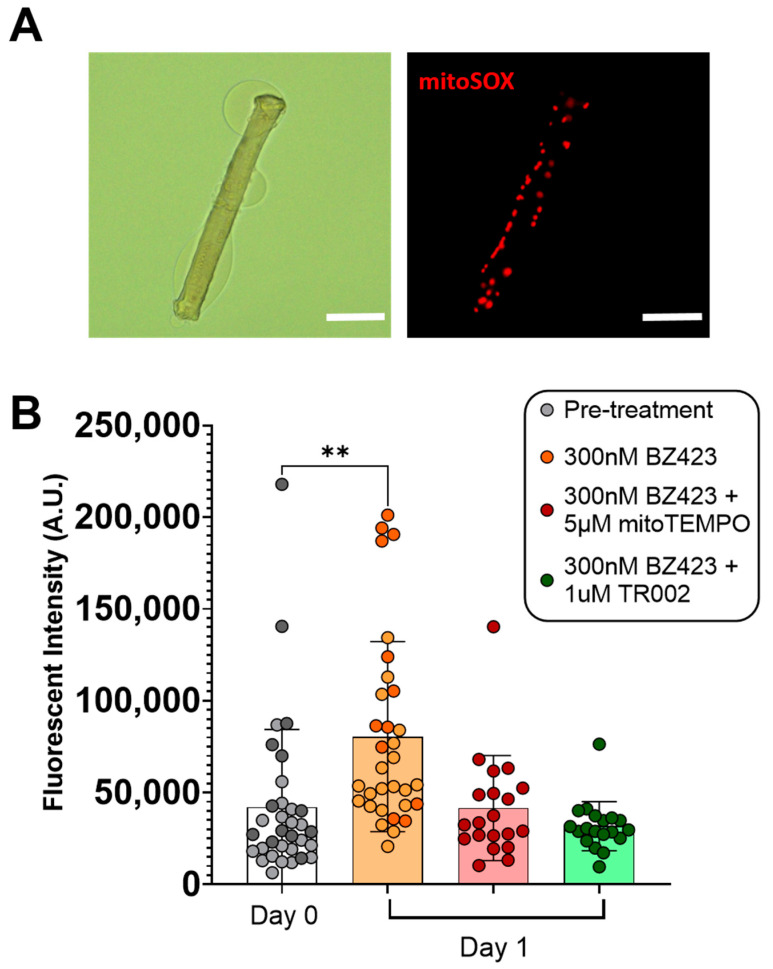
To examine the impact of chemical induction of MPT on mROS emission, single isolated myofibers were treated with either 300 nM Bz423, 300 nM BZ423 + 5 µM mitoTEMPO (to scavenge ROS), or 300 nM BZ423 + 1 µM TR002 (to inhibit MPT) for 24 h, and mitoSOX signal was determined at baseline and post-treatment. (**A**) Representative brightfield and fluorescence images of single FDB muscle fibers following 24 h treatment with Bz423. Scale bars represent 100 µm. (**B**) Both mitoTEMPO and TR002 prevented an increase in mitoSOX fluorescence over the 24 h treatment period, while fluorescence increased following treatment with BZ423, indicating an increase in mROS emission. Biological replicates are identified by data points of different colors within a given treatment, while technical replicates share the same color data point. *n* = 2 biological replicates were used for pre-treatment and 300 nM Bz423-treated myofibers, while *n* = 1 biological replicate was used for all other groups. ** *p* < 0.01 versus Day 0.

**Figure 3 cells-10-02586-f003:**
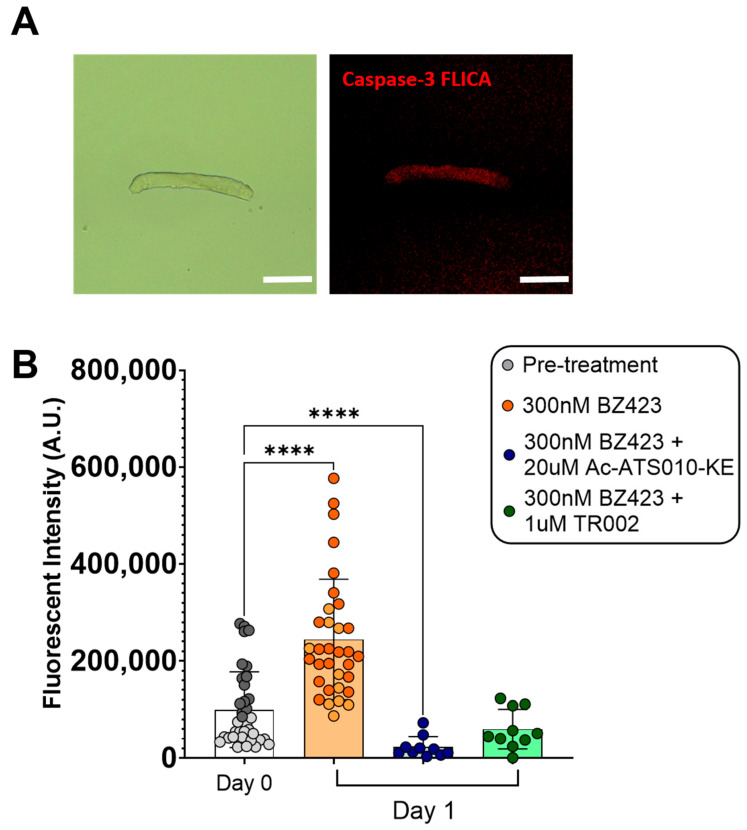
To examine the impact of chemical induction of MPT on caspase-3 activation, single isolated myofibers were treated with either 300 nM Bz423, 300 nM BZ423 + 20 µM Ac-ATS010-KE (to inhibit caspase-3), or 300 nM BZ423 + 1 µM TR002 (to inhibit MPT) for 24 h, and caspase-3 FLICA signal was determined at baseline and post-treatment. (**A**) Representative brightfield and fluorescence images of single FDB muscle fibers following 24 h treatment with Bz423. Scale bars represent 100 µm. (**B**) While TR002 prevented an increase in caspase-3 FLICA fluorescence over the 24 h treatment period, Ac-ATS010-KE decreased signal compared to Day 0. Treatment with BZ423 induced an increase in caspase-3 FLICA, indicating an increase in caspase-3 activity following treatment. Biological replicates are identified by data points of different colors within a given treatment, while technical replicates share the same color data point. *n* = 2 biological replicates were used for pre-treatment and 300 nM Bz423-treated myofibers, while *n* = 1 biological replicate was used for all other groups. **** *p* < 0.0001 versus Day 0.

**Figure 4 cells-10-02586-f004:**
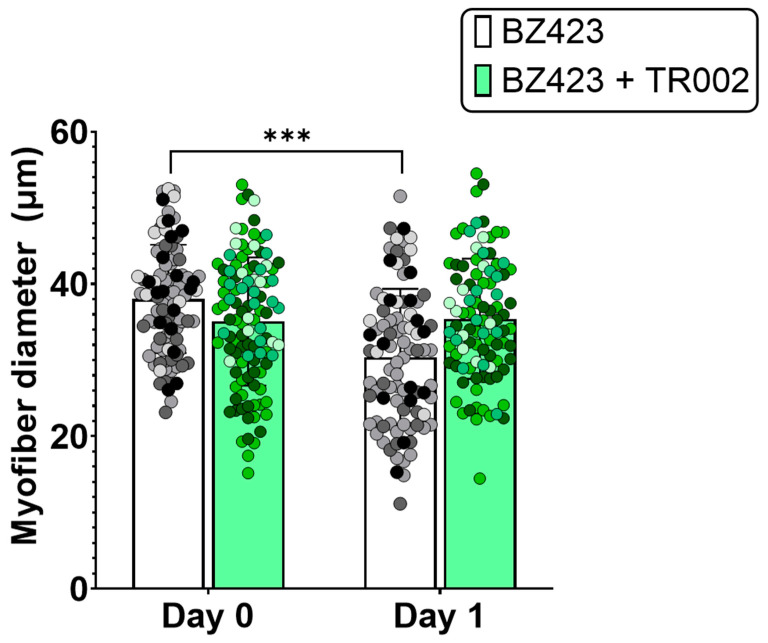
To determine if MPT causes atrophy in skeletal muscle fibers, single isolated mouse FDB myofibers were treated for 24 h with 300 nM Bz423 (MPT-inducer) or 300 nM Bz423 + 1 µM TR002 (MPT-inhibitor). Induction of MPT with Bz423 caused a decrease in myofiber diameter from pre- to post-treatment, whereas myofibers treated with the MPT inhibitor TR002 showed no change over this same period. Biological replicates (*n* = 4) are identified by data points of different colors within a given treatment, while technical replicates share the same color data point. *** *p* < 0.001 versus Day 0.

**Figure 5 cells-10-02586-f005:**
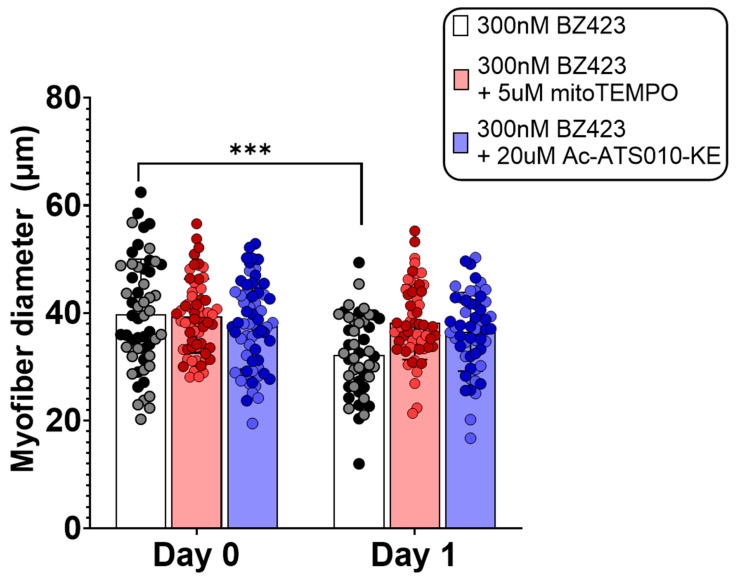
To determine if inhibition of mROS or caspase-3 prevents MPT-induced muscle fiber atrophy, isolated mouse FDB muscle fibers were treated with 300 nM Bz423 (MPT-inducer) or 300 nM Bz423 and either 5 µM mitoTEMPO (mROS-inhibitor) or 20 µM Ac-ATS010-KE (caspase-3-inhibitor). Both inhibition of mROS and caspase-3 prevented MPT-induced reduction in myofiber diameter. Biological replicates (*n* = 2) are identified by data points of different colors within a given treatment, while technical replicates share the same color data point. *** *p* < 0.001 versus Day 0.

**Figure 6 cells-10-02586-f006:**
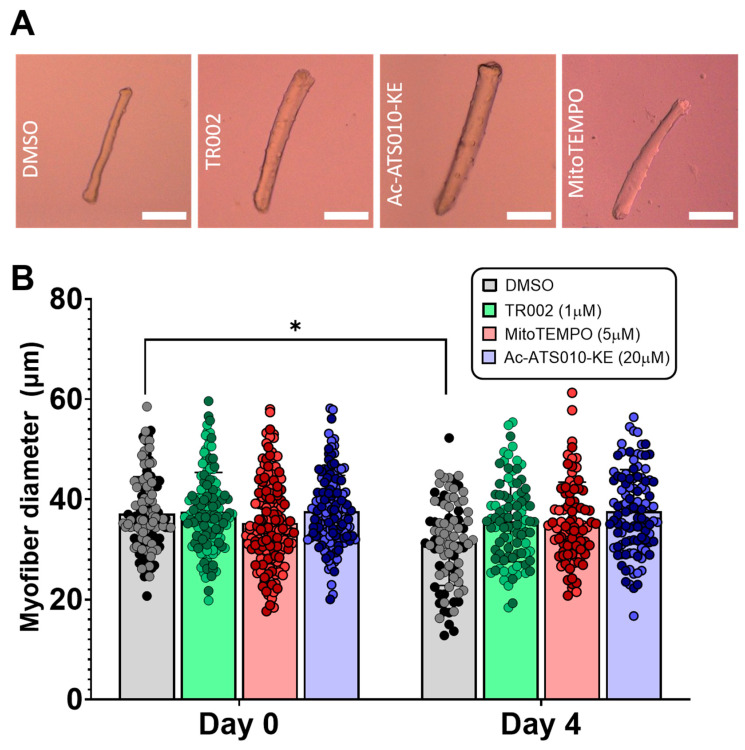
To determine if inhibition of MPT, mROS, or caspase-3 prevents atrophy in a single muscle fiber model of disuse atrophy, single isolated FDB myofibers were kept in culture conditions and treated with DMSO (vehicle control),1 µM TR002 (MPT-inhibitor), 20 µM Ac-ATS010-KE (caspase-3 inhibitor), or 5 µM mitoTEMPO (mROS-inhibitor) for 4 days, and myofiber diameter was assessed pre- and post-treatment. (**A**) Representative images of isolated mouse FDB fibers following 4-days of treatment with DMSO, TR002, Ac-ATS010, or mitoTEMPO. Scale bars represent 100 µm. Biological replicates (*n* = 2) are identified by data points of different colors within a given treatment, while technical replicates share the same color data point. (**B**) Inhibition of MPT, caspase-3, and mROS all prevented change in myofiber diameter over the 4-day period, while the vehicle control group demonstrated a substantial decline in size. This indicates that inhibition of MPT, caspase-3, or mROS prevent atrophy in a culture model of skeletal muscle disuse. * *p* < 0.05.

## Data Availability

Data and materials supporting the conclusions of this work are included herein. RTH is to be contacted to request availability of these materials.

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
