# Peer review of "Mitochondrial Permeability Transition Causes Mitochondrial Reactive Oxygen Species- and Caspase 3-Dependent Atrophy of Single Adult Mouse Skeletal Muscle Fibers"

_cells, 2021, doi:10.3390/cells10102586_

Round 1

Reviewer 1 Report

The aim of this study is to show MPT impacts skeletal muscle atrophy in isolated single mouse flexor digitorum brevis muscle fibers.

Generally, 300 nM of the benzodiazepine Bz423 are used to induce MPT, the generation of ROS and caspase 3 activation. Published literature (for example Giorgio et al 2013, Starke et al 2018) show an effect with 6-25 uM Bz423, which is significantly more than used in this study. The authors should at least discuss, why the used concentration of Bz423 is adequate for the performed experiments.  

Methods: Lines 117-119 say that “muscles were then triturated with a P1000 pipette for dissociation into single muscle cells fibers. Every 15 min, fibers were placed back at 37°C and 5% CO2 for 5 min to avoid substantial change in media temperature”. Please check the protocol, as it implies that trituration of the fibers into single cells takes 2 or more cycles of 15 minutes each, interrupted by 5 minutes of standard culture conditions in an incubator.

Line 123: Please check “5 mm culture dishes”? Are these wells of a 96 well plate?

Line 161: what is the freehand drawing tool? Please add missing information  

Figure 2C and 2D show quantification of the fluorescence of MitoSOX and FLICA in myofibers before and after treatment. The fluorescence of MitoSOX or FLICA increases during this time, however, the cells contract during this time too, as indicated in figures 2A and B. This makes it difficult to interpret the data. The same problem arises for Figure 3, 4 and 5, which show the change of the diameter of myofibers with respect to treatment with TR002, TEMPO or Ac-ATS010-KE. It is not clear to me how to distinguish between contraction of the myofibers and the change of diameter due to atrophy.

Figures 3, 4 and 5: the diameter of myofibers of multiple groups are compared, therefore ANOVA should be used to establish significance.

Figure 3: The terms biological and technical replicates should be more clearly defined. I assume that for this figure myofibers from 2 independent preparations were used (biological replicates), but multiple cells from each preparation were analyzed (technical replicates).  

Figure 3 shows values for each experiment in 2 different colors to indicate biological and technical replicates. Are the results in figures 4 and 5 just technical replicates?    

In addition the novel MPT inhibitor TR002 is used. Since it is not known, how TR002 inhibits MPT, the authors should add treatment with cyclosporine A to their experiments. Cyclophilin is a well-established regulator of the MPT and is inhibited by cyclosporine A. The mechanism of TR002 is not known yet. Structural TR002 shares some gross-similarity to ADP, which is also a known inhibitor of the MPT, but ADP has like TR002 no direct effect on the cyclophilin d activity.  

Caspase inhibitor: No control experiment (like a western blot showing the full length or cleaved caspase 3) is provided to show the effectiveness of the inhibitor

Line 37/38: Please add reference to mentioned studies

Line 57/58: Please add reference to mentioned studies

The aim of this study is to show MPT impacts skeletal muscle atrophy in isolated single mouse flexor digitorum brevis muscle fibers.

Generally, 300 nM of the benzodiazepine Bz423 are used to induce MPT, the generation of ROS and caspase 3 activation. Published literature (for example Giorgio et al 2013, Starke et al 2018) show an effect with 6-25 uM Bz423, which is significantly more than used in this study. The authors should at least discuss, why the used concentration of Bz423 is adequate for the performed experiments.  

Methods: Lines 117-119 say that “muscles were then triturated with a P1000 pipette for dissociation into single muscle cells fibers. Every 15 min, fibers were placed back at 37°C and 5% CO2 for 5 min to avoid substantial change in media temperature”. Please check the protocol, as it implies that trituration of the fibers into single cells takes 2 or more cycles of 15 minutes each, interrupted by 5 minutes of standard culture conditions in an incubator.

Line 123: Please check “5 mm culture dishes”? Are these wells of a 96 well plate?

Line 161: what is the freehand drawing tool? Please add missing information  

Figure 2C and 2D show quantification of the fluorescence of MitoSOX and FLICA in myofibers before and after treatment. The fluorescence of MitoSOX or FLICA increases during this time, however, the cells contract during this time too, as indicated in figures 2A and B. This makes it difficult to interpret the data. The same problem arises for Figure 3, 4 and 5, which show the change of the diameter of myofibers with respect to treatment with TR002, TEMPO or Ac-ATS010-KE. It is not clear to me how to distinguish between contraction of the myofibers and the change of diameter due to atrophy.

Figures 3, 4 and 5: the diameter of myofibers of multiple groups are compared, therefore ANOVA should be used to establish significance.

Figure 3: The terms biological and technical replicates should be more clearly defined. I assume that for this figure myofibers from 2 independent preparations were used (biological replicates), but multiple cells from each preparation were analyzed (technical replicates).  

Figure 3 shows values for each experiment in 2 different colors to indicate biological and technical replicates. Are the results in figures 4 and 5 just technical replicates?    

In addition the novel MPT inhibitor TR002 is used. Since it is not known, how TR002 inhibits MPT, the authors should add treatment with cyclosporine A to their experiments. Cyclophilin is a well-established regulator of the MPT and is inhibited by cyclosporine A. The mechanism of TR002 is not known yet. Structural TR002 shares some gross-similarity to ADP, which is also a known inhibitor of the MPT, but ADP has like TR002 no direct effect on the cyclophilin d activity.  

Caspase inhibitor: No control experiment (like a western blot showing the full length or cleaved caspase 3) is provided to show the effectiveness of the inhibitor

Line 37/38: Please add reference to mentioned studies

Line 57/58: Please add reference to mentioned studies

The aim of this study is to show MPT impacts skeletal muscle atrophy in isolated single mouse flexor digitorum brevis muscle fibers.

Generally, 300 nM of the benzodiazepine Bz423 are used to induce MPT, the generation of ROS and caspase 3 activation. Published literature (for example Giorgio et al 2013, Starke et al 2018) show an effect with 6-25 uM Bz423, which is significantly more than used in this study. The authors should at least discuss, why the used concentration of Bz423 is adequate for the performed experiments.  

Methods: Lines 117-119 say that “muscles were then triturated with a P1000 pipette for dissociation into single muscle cells fibers. Every 15 min, fibers were placed back at 37°C and 5% CO2 for 5 min to avoid substantial change in media temperature”. Please check the protocol, as it implies that trituration of the fibers into single cells takes 2 or more cycles of 15 minutes each, interrupted by 5 minutes of standard culture conditions in an incubator.

Line 123: Please check “5 mm culture dishes”? Are these wells of a 96 well plate?

Line 161: what is the freehand drawing tool? Please add missing information  

Figure 2C and 2D show quantification of the fluorescence of MitoSOX and FLICA in myofibers before and after treatment. The fluorescence of MitoSOX or FLICA increases during this time, however, the cells contract during this time too, as indicated in figures 2A and B. This makes it difficult to interpret the data. The same problem arises for Figure 3, 4 and 5, which show the change of the diameter of myofibers with respect to treatment with TR002, TEMPO or Ac-ATS010-KE. It is not clear to me how to distinguish between contraction of the myofibers and the change of diameter due to atrophy.

Figures 3, 4 and 5: the diameter of myofibers of multiple groups are compared, therefore ANOVA should be used to establish significance.

Figure 3: The terms biological and technical replicates should be more clearly defined. I assume that for this figure myofibers from 2 independent preparations were used (biological replicates), but multiple cells from each preparation were analyzed (technical replicates).  

Figure 3 shows values for each experiment in 2 different colors to indicate biological and technical replicates. Are the results in figures 4 and 5 just technical replicates?    

In addition the novel MPT inhibitor TR002 is used. Since it is not known, how TR002 inhibits MPT, the authors should add treatment with cyclosporine A to their experiments. Cyclophilin is a well-established regulator of the MPT and is inhibited by cyclosporine A. The mechanism of TR002 is not known yet. Structural TR002 shares some gross-similarity to ADP, which is also a known inhibitor of the MPT, but ADP has like TR002 no direct effect on the cyclophilin d activity.  

Caspase inhibitor: No control experiment (like a western blot showing the full length or cleaved caspase 3) is provided to show the effectiveness of the inhibitor

Line 37/38: Please add reference to mentioned studies

Line 57/58: Please add reference to mentioned studies

The aim of this study is to show MPT impacts skeletal muscle atrophy in isolated single mouse flexor digitorum brevis muscle fibers.

Generally, 300 nM of the benzodiazepine Bz423 are used to induce MPT, the generation of ROS and caspase 3 activation. Published literature (for example Giorgio et al 2013, Starke et al 2018) show an effect with 6-25 uM Bz423, which is significantly more than used in this study. The authors should at least discuss, why the used concentration of Bz423 is adequate for the performed experiments.  

Methods: Lines 117-119 say that “muscles were then triturated with a P1000 pipette for dissociation into single muscle cells fibers. Every 15 min, fibers were placed back at 37°C and 5% CO2 for 5 min to avoid substantial change in media temperature”. Please check the protocol, as it implies that trituration of the fibers into single cells takes 2 or more cycles of 15 minutes each, interrupted by 5 minutes of standard culture conditions in an incubator.

Line 123: Please check “5 mm culture dishes”? Are these wells of a 96 well plate?

Line 161: what is the freehand drawing tool? Please add missing information  

Figure 2C and 2D show quantification of the fluorescence of MitoSOX and FLICA in myofibers before and after treatment. The fluorescence of MitoSOX or FLICA increases during this time, however, the cells contract during this time too, as indicated in figures 2A and B. This makes it difficult to interpret the data. The same problem arises for Figure 3, 4 and 5, which show the change of the diameter of myofibers with respect to treatment with TR002, TEMPO or Ac-ATS010-KE. It is not clear to me how to distinguish between contraction of the myofibers and the change of diameter due to atrophy.

Figures 3, 4 and 5: the diameter of myofibers of multiple groups are compared, therefore ANOVA should be used to establish significance.

Figure 3: The terms biological and technical replicates should be more clearly defined. I assume that for this figure myofibers from 2 independent preparations were used (biological replicates), but multiple cells from each preparation were analyzed (technical replicates).  

Figure 3 shows values for each experiment in 2 different colors to indicate biological and technical replicates. Are the results in figures 4 and 5 just technical replicates?    

In addition the novel MPT inhibitor TR002 is used. Since it is not known, how TR002 inhibits MPT, the authors should add treatment with cyclosporine A to their experiments. Cyclophilin is a well-established regulator of the MPT and is inhibited by cyclosporine A. The mechanism of TR002 is not known yet. Structural TR002 shares some gross-similarity to ADP, which is also a known inhibitor of the MPT, but ADP has like TR002 no direct effect on the cyclophilin d activity.  

Caspase inhibitor: No control experiment (like a western blot showing the full length or cleaved caspase 3) is provided to show the effectiveness of the inhibitor

Line 37/38: Please add reference to mentioned studies

Line 57/58: Please add reference to mentioned studies

The aim of this study is to show MPT impacts skeletal muscle atrophy in isolated single mouse flexor digitorum brevis muscle fibers.

Generally, 300 nM of the benzodiazepine Bz423 are used to induce MPT, the generation of ROS and caspase 3 activation. Published literature (for example Giorgio et al 2013, Starke et al 2018) show an effect with 6-25 uM Bz423, which is significantly more than used in this study. The authors should at least discuss, why the used concentration of Bz423 is adequate for the performed experiments.  

Methods: Lines 117-119 say that “muscles were then triturated with a P1000 pipette for dissociation into single muscle cells fibers. Every 15 min, fibers were placed back at 37°C and 5% CO2 for 5 min to avoid substantial change in media temperature”. Please check the protocol, as it implies that trituration of the fibers into single cells takes 2 or more cycles of 15 minutes each, interrupted by 5 minutes of standard culture conditions in an incubator.

Line 123: Please check “5 mm culture dishes”? Are these wells of a 96 well plate?

Line 161: what is the freehand drawing tool? Please add missing information  

Figure 2C and 2D show quantification of the fluorescence of MitoSOX and FLICA in myofibers before and after treatment. The fluorescence of MitoSOX or FLICA increases during this time, however, the cells contract during this time too, as indicated in figures 2A and B. This makes it difficult to interpret the data. The same problem arises for Figure 3, 4 and 5, which show the change of the diameter of myofibers with respect to treatment with TR002, TEMPO or Ac-ATS010-KE. It is not clear to me how to distinguish between contraction of the myofibers and the change of diameter due to atrophy.

Figures 3, 4 and 5: the diameter of myofibers of multiple groups are compared, therefore ANOVA should be used to establish significance.

Figure 3: The terms biological and technical replicates should be more clearly defined. I assume that for this figure myofibers from 2 independent preparations were used (biological replicates), but multiple cells from each preparation were analyzed (technical replicates).  

Figure 3 shows values for each experiment in 2 different colors to indicate biological and technical replicates. Are the results in figures 4 and 5 just technical replicates?    

In addition the novel MPT inhibitor TR002 is used. Since it is not known, how TR002 inhibits MPT, the authors should add treatment with cyclosporine A to their experiments. Cyclophilin is a well-established regulator of the MPT and is inhibited by cyclosporine A. The mechanism of TR002 is not known yet. Structural TR002 shares some gross-similarity to ADP, which is also a known inhibitor of the MPT, but ADP has like TR002 no direct effect on the cyclophilin d activity.  

Caspase inhibitor: No control experiment (like a western blot showing the full length or cleaved caspase 3) is provided to show the effectiveness of the inhibitor

Line 37/38: Please add reference to mentioned studies

Line 57/58: Please add reference to mentioned studies

The aim of this study is to show MPT impacts skeletal muscle atrophy in isolated single mouse flexor digitorum brevis muscle fibers.

Generally, 300 nM of the benzodiazepine Bz423 are used to induce MPT, the generation of ROS and caspase 3 activation. Published literature (for example Giorgio et al 2013, Starke et al 2018) show an effect with 6-25 uM Bz423, which is significantly more than used in this study. The authors should at least discuss, why the used concentration of Bz423 is adequate for the performed experiments.  

Methods: Lines 117-119 say that “muscles were then triturated with a P1000 pipette for dissociation into single muscle cells fibers. Every 15 min, fibers were placed back at 37°C and 5% CO2 for 5 min to avoid substantial change in media temperature”. Please check the protocol, as it implies that trituration of the fibers into single cells takes 2 or more cycles of 15 minutes each, interrupted by 5 minutes of standard culture conditions in an incubator.

Line 123: Please check “5 mm culture dishes”? Are these wells of a 96 well plate?

Line 161: what is the freehand drawing tool? Please add missing information  

Figure 2C and 2D show quantification of the fluorescence of MitoSOX and FLICA in myofibers before and after treatment. The fluorescence of MitoSOX or FLICA increases during this time, however, the cells contract during this time too, as indicated in figures 2A and B. This makes it difficult to interpret the data. The same problem arises for Figure 3, 4 and 5, which show the change of the diameter of myofibers with respect to treatment with TR002, TEMPO or Ac-ATS010-KE. It is not clear to me how to distinguish between contraction of the myofibers and the change of diameter due to atrophy.

Figures 3, 4 and 5: the diameter of myofibers of multiple groups are compared, therefore ANOVA should be used to establish significance.

Figure 3: The terms biological and technical replicates should be more clearly defined. I assume that for this figure myofibers from 2 independent preparations were used (biological replicates), but multiple cells from each preparation were analyzed (technical replicates).  

Figure 3 shows values for each experiment in 2 different colors to indicate biological and technical replicates. Are the results in figures 4 and 5 just technical replicates?    

In addition the novel MPT inhibitor TR002 is used. Since it is not known, how TR002 inhibits MPT, the authors should add treatment with cyclosporine A to their experiments. Cyclophilin is a well-established regulator of the MPT and is inhibited by cyclosporine A. The mechanism of TR002 is not known yet. Structural TR002 shares some gross-similarity to ADP, which is also a known inhibitor of the MPT, but ADP has like TR002 no direct effect on the cyclophilin d activity.  

Caspase inhibitor: No control experiment (like a western blot showing the full length or cleaved caspase 3) is provided to show the effectiveness of the inhibitor

Line 37/38: Please add reference to mentioned studies

Line 57/58: Please add reference to mentioned studies

The aim of this study is to show MPT impacts skeletal muscle atrophy in isolated single mouse flexor digitorum brevis muscle fibers.

Generally, 300 nM of the benzodiazepine Bz423 are used to induce MPT, the generation of ROS and caspase 3 activation. Published literature (for example Giorgio et al 2013, Starke et al 2018) show an effect with 6-25 uM Bz423, which is significantly more than used in this study. The authors should at least discuss, why the used concentration of Bz423 is adequate for the performed experiments.  

Methods: Lines 117-119 say that “muscles were then triturated with a P1000 pipette for dissociation into single muscle cells fibers. Every 15 min, fibers were placed back at 37°C and 5% CO2 for 5 min to avoid substantial change in media temperature”. Please check the protocol, as it implies that trituration of the fibers into single cells takes 2 or more cycles of 15 minutes each, interrupted by 5 minutes of standard culture conditions in an incubator.

Line 123: Please check “5 mm culture dishes”? Are these wells of a 96 well plate?

Line 161: what is the freehand drawing tool? Please add missing information  

Figure 2C and 2D show quantification of the fluorescence of MitoSOX and FLICA in myofibers before and after treatment. The fluorescence of MitoSOX or FLICA increases during this time, however, the cells contract during this time too, as indicated in figures 2A and B. This makes it difficult to interpret the data. The same problem arises for Figure 3, 4 and 5, which show the change of the diameter of myofibers with respect to treatment with TR002, TEMPO or Ac-ATS010-KE. It is not clear to me how to distinguish between contraction of the myofibers and the change of diameter due to atrophy.

Figures 3, 4 and 5: the diameter of myofibers of multiple groups are compared, therefore ANOVA should be used to establish significance.

Figure 3: The terms biological and technical replicates should be more clearly defined. I assume that for this figure myofibers from 2 independent preparations were used (biological replicates), but multiple cells from each preparation were analyzed (technical replicates).  

Figure 3 shows values for each experiment in 2 different colors to indicate biological and technical replicates. Are the results in figures 4 and 5 just technical replicates?    

In addition the novel MPT inhibitor TR002 is used. Since it is not known, how TR002 inhibits MPT, the authors should add treatment with cyclosporine A to their experiments. Cyclophilin is a well-established regulator of the MPT and is inhibited by cyclosporine A. The mechanism of TR002 is not known yet. Structural TR002 shares some gross-similarity to ADP, which is also a known inhibitor of the MPT, but ADP has like TR002 no direct effect on the cyclophilin d activity.  

Caspase inhibitor: No control experiment (like a western blot showing the full length or cleaved caspase 3) is provided to show the effectiveness of the inhibitor

Line 37/38: Please add reference to mentioned studies

Line 57/58: Please add reference to mentioned studies

Reviewer 2 Report

Cells (cells-1363824), Comments to the Authors:

Title: Mitochondrial permeability transition causes mitochondrial reactive oxygen species- and caspase 3-dependent atrophy of single adult mouse skeletal muscle fibers

Comments:

The submitted manuscript discussed the effect of mitochondrial permeability transition (MPT) on mROS emission and activation of caspase-3 via mitochondrial release of cytochrome c. The authors tested MPT as a mechanism of atrophy. Treating cultured single mouse flexor digitorum brevis (FDB) fibers from adult mice with a chemical inducer of MPT (Bz423) for 24 h caused an increase in mROS and caspase-3 activity that was accompanied by a reduction in muscle fiber diameter that was able to be prevented by inhibitors of MPT, mROS or caspase-3 (P<0.05). Similarly, 4-day single fiber culture as a model of disuse caused atrophy that could be prevented by inhibitors of MPT, mROS, or activated caspase-3.

I think the submitted manuscript can be accepted after the authors respond to the following comments:

  1. The following 2 sentences contradict each other “Despite MPT being associated with both an increase in mROS 54 and caspase-3 activity, prior studies have not considered the possibility of a role for MPT 55 as a cause of skeletal muscle atrophy” and “Consistent with the idea that MPT may be involved in muscle atrophy, several studies have provided evidence of MPT in skeletal muscle in conditions associated with atrophy.”
  2. Figure 1 quality is poor. The authors should try to draw a more professional diagram to summarize their hypothesis.
  3. The scale bar should be placed in the photos depicting the muscle cell.
  4. The authors should indicate in Figure 5A the treatment on each photo of muscle fibers.
  5. The authors should discuss the applications of their findings.

Round 2

Reviewer 1 Report

I have no further concerns.

Reviewer 2 Report

Cells (cells-1363824), Comments to the Authors:

Title: Mitochondrial permeability transition causes mitochondrial reactive oxygen species- and caspase 3-dependent atrophy of single adult mouse skeletal muscle fibers

Comments:

After reading the authors' response to my comments, I think the manuscript can be accepted for publication.